# Detecting Changes in Forced Climate Attractors with Wasserstein Distance

Yoann Robin[1], Pascal Yiou[1], and Philippe Naveau[1]

[1]LSCE, Gif-sur-Yvette, France

*Correspondence to:* Y. Robin (yoann.robin@lsce.ipsl.fr)

**Abstract.**

## 1 Introduction

If the climate system is viewed as a complex dynamical system yielding a strange attractor, i.e. a highly complicated object around which all trajectories wind up (Lorenz, 1963), then, climate variability is linked to the statistical properties of such an attractor (Ghil and Childress, 1987). Those statistical properties refer to the probability that trajectories visit each region of phase space (Mané, 2012; Eckmann and Ruelle, 1985). Mathematical concepts to describe those properties on rather simple dynamical systems have been investigated by Chekroun et al. (2011).

In addition to climate internal variability, external forcings (either natural or anthropogenic) perturb the climate system dynamics by introducing a time dependence of the attractor. This is the cause of non-stationary behavior of the climate system. At first order, this can translate into a general shift of the underlying attractor (Corti S. et al., 1999). At second order, interactions between a seasonal cycle and a slow forcing can even lead to trends in subtle quantities (e.g. Cassou and Cattiaux, 2016; Vrac et al., 2014). A few properties of the climate attractor due to external forcings (anthropogenic or not) have been treated by Pierini et al. (2016) and Drótos et al. (2015), who focused on low dimensional strange attractors and investigated qualitative changes of the attractors, although all those studies are quantitative in many aspects. Lucarini, V. et al. (2017) have recently used response theory (Ruelle, D., 2009) to quantify the modification of the dynamics submitted to a forcing.

Classical distances, like the Euclidean distance are often used to measure attractor differences. The goal of our paper is to present a framework, embedded in optimal transport theory (e.g. Villani, 2003), to measure the distance between strange attractors, and make a statistical inference of this tool on well documented dynamical systems. To do this we exploit the fact that the attractor of the system defines an invariant measure, which is the multivariate probability distribution of all trajectories of the system. The distance between attractors is then computed through the cost to transform one invariant measure into another. A similar idea was already proposed in Ghil (2015) to characterize the climate variability. In particular, we assess that it is possible to discriminate between attractors, given a relatively low number of sampling points, in order to ensure the applicability of this methodology. We test this method on a time-varying dynamical system in order to illustrate how the dynamics of a system can be affected by a constant forcing interacting with seasonality.

25      The paper is organized as follows. In Section 2, we recall some basic concepts used in optimal transport theory and recall the definition of the Wasserstein distance. In Section 3, we investigate the performance of the Wasserstein distance to discriminate between two "simple" autonomous systems (winter against summer of Lorenz (1984) model). Section 4 explores how forcing can impact the Wasserstein distance capability at detecting changes in a non stationarity context. Section 5 concludes and proposes some future research directions.

## 2  Distance between measures

To characterize changes in the properties of the attractor of a dynamical system, the first step of our methodology is to determine how two measures (or distributions of mass) differ. The idea is to derive a cost function for transporting one mass distribution onto the other. As a simple example, we consider the three mass distributions shown in Figure 1, noted $\mu$, $\nu$ and $\xi$. The distributions are on a grid of size $10 \times 10 = 100$, with mass positions located on pixels $\mathbf{x}_i$, $i = 1, \ldots, 100$. $\nu$ is constructed to
be a one pixel left shift of $\mu$. The distribution $\xi$ is a $90°$ rotation and a mirror image of $\mu$, and we move one square to have a common point with $\mu$. The distribution $\mu$ (resp. $\nu$ and $\xi$) can be written as

$$\mu = \sum_{i=1}^{100} \mu_i \delta_{\mathbf{x}_i},$$

where $\delta_{\mathbf{x}}$ is the Dirac mass at pixel $\mathbf{x}_i$, and $\mu_i = 1$ on the black boxes in Figure 1 and $\mu_i = 0$ on the grey boxes. The Euclidean distance $\mathbf{d}$ between $\mu$ and $\nu$ is defined by

$$15 \quad \mathbf{d}(\mu, \nu)^2 = \sum_{i=1}^{100} |\mu_i - \nu_i|^2.$$

Panels a and b in Figure 1 are visually very similar, whereas Panel c cannot be deduced from a trivial transformation of the first panels. Therefore, it is expected that $\mu$ is "closer" to $\nu$ than $\xi$. We find that the Euclidean distance from $\mu$ to $\nu$ is 3.74, and the distance from $\mu$ to $\xi$ is 3.46 (the example was constructed to show this). Thus the Euclidean distance does not capture the structural proximity between $\mu$ and $\nu$. The explanation is the following: among the squares that have no common mass, the
value of the Euclidean distance is independent of the position of squares. This highlights the need of a distance that can take into account how masses should be moved, say, from the left panel to the middle panel of Figure 1.

     This mathematical problem traces back to Monge (1781) and is the basis of optimal transport theory (see, e.g. Villani, 2003). To transport the mass distribution $\mu$ contained in the boxes at $\mathbf{x}_i$ to the distribution $\nu$ in the boxes at $\mathbf{x}_j$, a total cost of the transport has to be defined. We note $\gamma_{ij} > 0$ the fraction of the mass transported from the boxes $\mathbf{x}_i$ to $\mathbf{x}_j$. The cost of the transport is defined by $\gamma_{ij} \mathbf{d}(\mathbf{x}_i, \mathbf{x}_j)^2$. Consequently, the total transport cost from $\mu$ to $\nu$ is

$$\sum_{ij} \gamma_{ij} \mathbf{d}(\mathbf{x}_i, \mathbf{x}_j)^2,$$

where $\mathbf{d}$ is the usual Euclidean distance between the location $\mathbf{x}_i$ and $\mathbf{x}_j$. The set of $\gamma_{ij}$ coefficients is called the *transport plan*. It is a measure on product space of measures admitting $\mu$ and $\nu$ as margins. The optimal transport cost is obtained by minimizing this sum over all possible transport plans, i.e. all possible $\gamma_{ij} > 0$. This produces the so-called Wasserstein distance

$$\mathcal{W}(\mu, \nu) = \left( \inf_{\gamma_{ij}} \sum_{ij} \gamma_{ij} \mathbf{d}(\mathbf{x}_i, \mathbf{x}_j)^2 \right)^{1/2}. \tag{1}$$

Computing the right hand side of Eq. (1) is a problem of minimization under constraints on the $\gamma_{ij}$ coefficients, which have to be positive, and whose marginal sums equal $\mu_i$ and $\nu_j$. This distance can be numerically computed by network simplex algorithms, coming from linear programming theory (see, e.g. Bazaraa et al., 2009). We refer to Appendix A for a general idea of the algorithm. Eq. (1) is the discrete version of a more general formulation of the Wasserstein distance whose properties are detailed by Villani (2003).

In our example (Fig. 1), we have $\mathcal{W}(\mu, \nu) = 1 \ll 3.27 = \mathcal{W}(\mu, \xi)$. Therefore we can quantify with the Wasserstein distance that the cost of transforming $\mu$ into $\nu$ is lower than transforming $\mu$ into $\xi$. This result is closer to the physical intuition that a small shift is less costly than a mirror image and a rotation. Our next step is to apply the Wasserstein distance to differentiate between dynamical systems.

## 3 Inference on simple dynamical systems

### 3.1 Attractors and measure of a dynamical system

A dynamical system can be defined by the action of an ordinary differential equation

$$\frac{\mathrm{d}\mathbf{x}}{\mathrm{d}t} = \mathbf{v}(\mathbf{x}),$$

on a set of initial conditions (see, e.g. Guckenheimer and Holmes, 1983; Katok and Hasselblatt, 1997). Here $\mathbf{x}$ is a multivariate vector in the so-called *phase space* and $\mathbf{v}(\mathbf{x})$ is a vector field that acts on $\mathbf{x}$. The properties of the ensemble of trajectories from all initial conditions define the *dynamics* of the system. They are entirely determined by $\mathbf{v}$.

For chaotic dynamical systems, trajectories $\mathbf{x}(t)$ emerging from almost all initial conditions converge to a unique object called an *attractor*, embedded in the phase space. Attractors define an *invariant measure* in phase space, which quantify the weight of all trajectories of the dynamical system in subregions of the phase space. The measure of a sub region of phase space is the probability of a trajectory of the system to go through the region. The invariance is characterized by the conservation of the volume by the dynamics of the system (Ruelle, 1989). The goal of this section is to estimate the distance between the empirical invariant measure of attractors in particular setup.

We now focus on the Lorenz (1984) model, which is an idealized model of the Hadley circulation and its seasonality. The dynamics of this system is noted $\mathbf{v}(\mathbf{x})$, and, for a vector $\mathbf{x} = (x_1, x_2, x_3)$ given by

$$\mathbf{v}(\mathbf{x}) = \begin{pmatrix} -x_2^2 - x_3^2 - (x_1 - F)/4 \\ x_1 x_2 - 4 x_1 x_3 - x_2 + 1 \\ x_1 x_3 + 4 x_1 x_2 - x_3 \end{pmatrix}. \tag{2}$$

We propose to discriminate two attractors based on Eq. (2). A first attractor is generated with $F \equiv 11.5$ (noted Wi, for winter). A second attractor is generated with $F \equiv 7.5$ (noted Su, for summer). We choose those values and this terminology because $F$ is interpreted as a seasonal cycle in Section 4, of length $\tau = 73$ units. Both systems have three variables (so the phase space is $\mathbb{R}^3$), are chaotic and yield a strange attractor. They are illustrated by two long trajectories in Figure 2. To quantify the difference between the two attractors, it is first necessary to estimate the invariant measure of both attractors. We use the method of *snapshot attractors* (e.g. Romeiras et al., 1990; Chekroun et al., 2011) rather than considering one single long trajectory that could bias the sampling of some regions of the attractors, and requires the system to be ergodic. In the snapshot attractors, we draw $N$ random initial conditions following a uniform distribution. All margins are independent. This approximates a Lebesgue measure in a cube that includes the attractors. We iterate the dynamics of the systems between $t_0 = 0$ and a long time multiple of $\tau$. Consistently with Drótos et al. (2015), we take $5\tau = 5 \times 73$ (i.e. 5 cycles, but we have checked than $\tau$ is enough). Both systems are dissipative outside of the attractors neighborhood, therefore all $N$ trajectories collapse to the attractors after time $5\tau$ and provide an efficient sampling of the invariant measure (Romeiras et al., 1990). After time $5\tau$, the set of $N$ final points emerging of $N$ initial conditions is called a *snapshot attractor* (see Algorithm 1). Snapshot attractors are special cases of *pullback attractors* (Chekroun et al., 2011). The latter class requires an integration between $-\infty$ and a desired final time. Eq. (2) does not depend of time, so the integration into Sec. 3.1 can be performed on any length intervals.

---

**Algorithm 1** Simulation of a snapshot attractor with $N$ initial conditions from the Lorenz 84 system

---

**Require:** $5\tau \ (= 5 \times 73)$ iteration time for convergence towards the attractor,

  $N \ (= 50, 100, 1000)$ the number of points in the snapshot,

  $C \ (= [-1, 3] \times [-3, 3] \times [-3, 3])$ a box that contains the attractor

---

**Ensure:** One snapshot with $N$ points denoted $\{\mathbf{y}^i\} \in \mathbb{R}^3$ with $i = 1, \ldots, N$

---

1: Draw uniformly $N$ points $\mathbf{x}^1, \ldots, \mathbf{x}^N$ in $C$

2: **for** $i = 1, \ldots, N$ **do**

3:   Integrate Eq. (2) between 0 and $5\tau$ starting to $\mathbf{x}^i$. The ending point is $\mathbf{y}^i$. Integration is performed using the RK4 scheme with a time step of 0.005.

4: **end for**

---

Then we compute the empirical measures associated with the snapshot attractors by discretizing the phase space (approxi-
mated by the box $[-1, 3] \times [-3, 3] \times [-3, 3]$) into cells of size $0.1 \times 0.1 \times 0.1$ (so $40 \times 60 \times 60 = A$ cells), and by counting the
number of points of a snapshot attractor in each cell (see Algorithm 2). The empirical measure of the winter attractor (resp. sum-
mer) is noted $\mu^{\text{Wi}}$ (resp. $\mu^{\text{Su}}$). They are sums of Dirac measures at each discrete cell. It is the equivalent of a multi-dimensional
histogram of the attractor. We chose a bin length of $0.1$ for the Lorenz attractor, which remains in a $[-1; 3] \times [-3; 3] \times [-3; 3]$
box. Therefore $40 \times 60 \times 60$ bins cover the attractor. This number of bins is comparable to the number of gridcells that cover
the North Atlantic region in the NCEP reanalysis (or most CMIP5 model simulations). This example refers to a few papers
dealing with climate attractor properties (e.g. Corti S. et al. (1999); Faranda D. et al. (2017)).

---

**Algorithm 2** Determining the empirical invariant measure from simulated snapshot attractors

---

**Require:** One snapshot attractor, $\{\mathbf{y}^i\}_{i=1,\dots,N}$, obtained from Algorithm 1,

    $[-1, 3] \times [-3, 3] \times [-3, 3]$, a large box containing the attractor,

    $0.1$, the length of the edge of each cells to compute the histogram (so $40 \times 60 \times 60 = A$ cells)

---

**Ensure:** An approximated density measure, i.e. a sum of Dirac masses estimated from the number of points in each cell $B_a$ with $a = 1, \dots, A$

$$\mu = \frac{1}{N} \sum_{a=1}^{A} \mu_a \delta_{B_a}$$

where $\delta_{B_a}$ is the Dirac measure around the cell $B_a$ (equal to one if $\mathbf{x} \in B_a$ and zero otherwise) and $\mu_a \geq 0$ is the inferred mass. $\mu_a$ is
not equal to $0$ for a small numbers of boxes.

---

1: Divide the space into small gridded cell $B_a$ of size $0.1 \times 0.1 \times 0.1$.
2: **for all** cells $B_a$ **do**
3:    $\mu_a \leftarrow$ (number of $\mathbf{y}^i$ in $B_a$)/$N$
4: **end for**

---

### 3.2 Protocol

The difference between the summer and winter attractors is evaluated by $\mathcal{W}\mu^{\text{Wi}}, \mu^{\text{Su}})$ for different sample sample of size $N$.
The probability distribution of Wasserstein distances is not known a priori for random measures. We first estimate the typical
value of Wasserstein distances between identical attractors in order to build a null hypothesis to be rejected if the distance is
larger to some threshold. Therefore, we construct fifty winter (resp. summer) Lorenz 84 snapshot attractors, with empirical
measure $\mu_k^{\text{Wi}}$, $k = 1, \dots, 50$ (resp. $\mu_k^{\text{Su}}$), by drawing fifty sets of $N$ random initial conditions, and applying Algorithms 1 and 2
between $0$ and $5\tau$. By construction, $\mathcal{W}(\mu_k^{\text{Wi}}, \mu_{\tilde{k}}^{\text{Wi}})$ should tend to $0$ when $N$ increase.

We detect a difference between the winter and summer of Lorenz 84 systems if

$$\mathcal{W}(\mu_k^{\text{Wi}}, \mu_{\tilde{k}}^{\text{Wi}}) \ll \mathcal{W}(\mu_k^{\text{Wi}}, \mu_{\tilde{k}}^{\text{Su}}) \text{ and } \mathcal{W}(\mu_k^{\text{Su}}, \mu_{\tilde{k}}^{\text{Su}}) \ll \mathcal{W}(\mu_k^{\text{Wi}}, \mu_{\tilde{k}}^{\text{Su}}).$$

This is quantified by a Kolmogorov-Smirnov (K.S.) test (Durbin, 1973; von Storch and Zwiers, 2001) between the distributions of $\mathcal{W}(\mu_k^{\mathrm{Wi}}, \mu_{\tilde{k}}^{\mathrm{Wi}})$ (resp. $\mathcal{W}(\mu_k^{\mathrm{Su}}, \mu_{\tilde{k}}^{\mathrm{Su}})$) and $\mathcal{W}(\mu_k^{\mathrm{Wi}}, \mu_{\tilde{k}}^{\mathrm{Su}})$, in order to reject the null hypothesis that the probability distributions are equal. The K.S. test gives two values, the maximal difference between the cumulative distribution function of measures, and the "$p$ value", which quantifies the probability to accept the null hypothesis. It is estimated by the Kolmogorov distribution (see (Marsaglia G. et al., 2003)). We choose to simulate 50 attractors of winter and 50 attractors of summer. We have $50 \times 50 = 2500$ different pairs between summer and winter. For the distances between the 50 attractors of the same season (summers or winters), we only consider $1 \leq (k, k') \leq 50$ pairs with $k < k'$. This means that we have 1225 distances for the winter or the summer. So we have at least 1000 distances per distribution. This is a reasonable sample size for a representative Kolmogorov-Smirnov test.

The estimation of the Wasserstein distance between attractors obviously depends on the number of available samples $N$ of the dynamical systems on which the empirical measures are constructed. To explore the variability in the estimation of Wasserstein distances from finite observational sets, we sample its distances for three different values of $N$: $N = 50$, 100 and 1000. We compute also one of each distance for $N = 10^6$. This later case represents a quasi-perfect estimation of the distance and we consider it as our benchmark for comparison.

The complete procedure to obtain an empirical probability distribution of Wasserstein distances, depending on the sample size $N$, is summarized in Algorithm 3.

---

**Algorithm 3** Estimation of $\mathcal{W}(\mu_k^{\mathrm{Wi}}, \mu_{\tilde{k}}^{\mathrm{Wi}})$, $\mathcal{W}(\mu_k^{\mathrm{Wi}}, \mu_{\tilde{k}}^{\mathrm{Su}})$ and $\mathcal{W}(\mu_k^{\mathrm{Wi}}, \mu_{\tilde{k}}^{\mathrm{Su}})$

---

**Require:** $N\ (= 50, 100, 1000)$ the number of points in snapshots

---

**Ensure:** 1225 independent estimates of the two Wasserstein distances $\mathcal{W}(\mu^{\mathrm{Wi}}, \mu^{\mathrm{Wi}})$ and $\mathcal{W}(\mu^{\mathrm{Su}}, \mu^{\mathrm{Su}})$ where the first differentiates two winter and the second two summer of Lorenz 84 snapshot attractors.

2500 independent estimates of the Wasserstein distances $\mathcal{W}(\mu^{\mathrm{Wi}}, \mu^{\mathrm{Su}})$ which compares winter and summer of Lorenz84 snapshot attractors.

---

1: Use Algorithm 1 to simulate fifty winter and fifty summer Lorenz 84 snapshot attractors, denoted $\mathrm{Wi}_k$ and $\mathrm{Su}_k$. Each snapshot attractor has $N$ points.

2: Use Algorithm 2 to transform each $\mathrm{Wi}_k$ (resp. $\mathrm{Su}_k$) into measures, noted $\mu_k^{\mathrm{Wi}}$ (resp. $\mu_k^{\mathrm{Su}}$).

3: Compute the Wasserstein distances $\mathcal{W}(\mu_k^{\mathrm{Wi}}, \mu_{\tilde{k}}^{\mathrm{Wi}})$ (resp. $\mathcal{W}(\mu_k^{\mathrm{Su}}, \mu_{\tilde{k}}^{\mathrm{Su}})$) for $k \neq \tilde{k}$ (see Appendix A). Thus, $\frac{50 \times (50+1)}{2} - 50 = 1225$ distances are stored.

4: Compute the Wasserstein distances $\mathcal{W}(\mu_k^{\mathrm{Wi}}, \mu_{\tilde{k}}^{\mathrm{Su}})$ for all $k, \tilde{k}$. Thus, $50^2 = 2500$ distances are stored.

---

### 3.3 Estimation

The probability distributions of the Wasserstein distance for $\mathcal{W}(\mu^{\mathrm{Wi}}, \mu^{\mathrm{Wi}})$ (resp. Su) and $\mathcal{W}(\mu^{\mathrm{Wi}}, \mu^{\mathrm{Su}})$ are summarized in Figure 3(a) by box-and-whisker plots (boxplots: Chambers et al., 1983) The distribution of the distances between winter (resp.

summer) snapshot attractors decreases to 0 (the expected asymptotic value) when $N$ increases (white and grey boxplots). We explain the relatively high values of the distance when $N = 50$ by the fact that few cells of the discrete measure are filled when $N$ is low, so that the transport plan is not zero. By increasing $N$, all cells tend to be sampled, so that the transport plans are less affected by sampling issues, and the cost of the transport decreases on average.

The distance between winter and summer attractors (black boxplots) decrease with $N$ and converge to the "true" value that is estimated with $N = 10^6$. The explanation is similar: if the measures of the snapshot attractors are estimated with low $N$, the "circles" composing the attractors are akin. Increasing the number of initial conditions $N$ essentially allows to differentiate the dynamics of the two attractors. We note that the distribution of distances for $N = 1000$ is very close to the one with $10^6$. This highlights a rather quick convergence of the Wasserstein distance $\mathcal{W}$. Figure 3(a) shows a good discrimination between null hypothesis distances $\mathcal{W}(\mu^{\mathrm{Wi}}, \mu^{\mathrm{Wi}})$ and the distances $\mathcal{W}(\mu^{\mathrm{Wi}}, \mu^{\mathrm{Su}})$, even for $N = 50$. This discrimination is confirmed by Kolmogorov-Smirnov tests reported in Table 1. The null hypothesis of identical attractors is rejected with probability one, even for $N = 50$. Finally, the variance of distance decrease with $N$. Indeed, attractors are independant of the initial condition, thus the variability is due to a low $N$. This propertie is shown by the Wasserstein distance.

This protocol was also applied for bin sizes of 0.05, 0.2 and 1.0. For 0.05 and 0.2, the maximal variation of median (resp. standard deviation) of Wasserstein distances is 0.03 (resp. 0.01), so we have the distributions are indistinguishable in practice. For a bin size of 1.0, the maximal increase of the median is 0.22, but the difference with the median of winter against summer is at least equal to 0.3.

For illustration purposes, we compute Euclidean distances between the same snapshot attractors (Figure 3(b)). The distances are normalized by $\sqrt{2}$, the maximum value being reached for two measures without common points. The distances between winter (resp. summer) snapshot attractors decrease as $N$ increase (white and grey boxplots). The distances between winter and summer snapshot attractors also decrease to the "perfect" estimate with $N = 10^6$ (black boxplots), but the convergence to the limit is far from being reached with $N = 1000$. The difference between winter and summer cannot be detected for all $N$. For $N = 50$ and 100, the distances between winter are greater than distances between winter and summer. For $N = 1000$, the Kolmogorov-Smirnov test (Table 1) shows the impossibility to reject the null hypothesis without ambiguity. Moreover, the variability is small and constant with $N$, which is incompatible with the high variability due to a low $N$. Therefore, the Euclidean distance might not be very useful to distinguish the dynamics in real world systems with few observations.

## 3.4 Inference with reduced information

In this section, we test whether it is possible to differentiate between attractors if only partial information is available. This can happen if one or more variables of the system are omitted (projection onto the remaining variables) or if variables are censored (truncation of the values of a variable), or a combination of both. The motivation in atmospheric sciences is that the underlying dynamical system is defined in three spatial dimensions (on the sphere), and that observables of the attractor of this system are generally obtained over a limited area (censoring of the rest of the globe) and a fixed pressure level (projection).

It has been proven that a sequence of *observables* of a dynamical system convey the same dynamics as the whole system (Packard et al., 1980; Takens, 1981; Mañé, 1981). Therefore it is meaningful to compare the distances between projected or truncated attractors.

For the Lorenz 84 attractors, a first reduction of information is performed by projecting the systems onto their $(x_1, x_2)$ variables (design P). We hence compute the distances (Wasserstein and Euclidean) between attractors from the variables $(x_1, x_2)$ and discard the information on $x_3$. The second reduction consists in truncating negative values of the variable $x_1$ (design T). Thus, we only consider the values of $(x_1, x_2, x_3)$ when $x_1 \geq 1$. The third reduction of information is a combination of projection onto the $(x_1, x_2)$ variables and truncating negative values of $x_1$ (design T+P). These transformation are illustrated in Figure 4(a-c). Those three transformations create observables of the underlying attractors. We shall call them "observed attractors", with designs P (projection), T (truncation) and P+T (both). The distribution of the distance between observed attractors is shown in Figure 4(d-i), for the two distances and each of the information reduction design (P, T and P+T).

The Wasserstein distance distribution (Figure 4(a-c)) shows a clear discrimination between winter and summer observed attractors, for all values of $N$. This is reflected in the Kolmogorov-Smirnov test: all test values are greater than $0.97$, except for winter with $N = 50$, this is $0.84$. All $p$-value are equal to $0$. The estimated distances between winter and summer observed attractors is always smaller than the idealized one (obtained on the full attractors) and shows little dependence on the number of points $N$. This is explained by the fact that the projection on a subspace of dimension 2 implies a reduction of transportation cost. Moreover, some points that are very far in the winter attractor become close to each other in the projection P. Overall, the reduction of information decreases the discriminating power of the Wasserstein distance, but the results are still significant for number of points $N$ as small as $50$.

The same experiment is conducted with the Euclidean distance (Figure 4(g-i)). Contradicting the intuition, it clearly discriminate between winter and summer for all designs P, T and P+T. Comparing the full attractors (Figure 2) and figures 4(a-c), we see that some points very far, become close in the same boxes of the estimated measure. This is reflected by a gain of variance, which decrease with $N$. Finally, we need $N = 1000$ to have the distribution between winters (resp. summers) lower than the idealized distance.

We conclude that the Wasserstein distance has a high capacity of discriminating different attractors coming from this dynamical system, even with a partial information. It is particularly promising in atmospheric sciences, where analyses are performed on truncated variables (e.g. a surface field on a limited area: transformation T) and/or on only one atmospheric field (e.g. geopotential height, omitting other variables: transformation P).

## 4   Time-varying dynamical system

We now focus on a time varying dynamical system that mimics variability around a seasonal cycle, and a monotonic forcing that plays after a triggering time $T$. Such a system defines a *snapshot* attractor at all times $t$. We want to measure how snapshot

attractors evolve after time $T$, when the forcing increases (we mean the forcing modifies more and more the attractors). The constant $F$ in the System 2 is now a function of time, and include a seasonal cycle and a forcing.

$$F(t) = 9.5 + \underbrace{2\sin\left(\frac{2\pi t}{73}\right)}_{\text{seasonality}} - \underbrace{2\frac{t-T}{T}\mathbf{1}_{\{t>T\}}}_{\text{monotonic forcing}}, \quad T = 100 \times 73. \tag{3}$$

The snapshot attractors of this system were investigated by Drótos et al. (2015), who performed an analysis of the mean and variance of each coordinate to detect the forcing $F$ after time $t > T$.

Such a coupled behaviour is present in most regional temperature time series at the decadal or centennial scales. The periodic part of the forcing $F$ in Eq. (3) allows one to divide the year into seasons of the system (Lorenz, 1984; Drótos et al., 2015). To be

consistent with Lorenz (1984) and Drótos et al. (2015), we consider that there are $73 = \tau$ time units in one year. We emphasize that a time unit is not analogous to a "day", but corresponds to a typical variability time scale in the non forced chaotic system in Eq. (2). We follow Drótos et al. (2015) and define the Fall equinox at $t = 0$ year or $t \mod 73 = 0$ year. Then, winter solstices correspond to $t \mod 73 = 0.25$ year, Spring equinoxes correspond to $t \mod 73 = 0.5$ year and summer solstices correspond to $t \mod 73 = 0.75$ year. This time dependent system produces a different snapshot attractor at each time step. We focus on

the snapshot attractors that occur at each equinoxe/solstice. These parameters are coherent with winter and summer defined in Section 3.

In this section, we want to quantify the change of the whole dynamics of the ensemble of snapshot attractors with the Wasserstein distance, and assess the detectability of changes from small numbers of observations.

## 4.1  Protocol

We compute snapshot attractors for each time step, for $t$ between 0 to 200 years. To have the convergence of trajectories on attractor, we draw $N$ initial conditions in a cube (see Algorithm 1) and perform a first integration during $5\tau = 5 \times 73$ time unit (i.e. 5 years). The attractor obtained is considered at $t = 0$. As previous Section, $N = 50, 100, 1000$. We generate also a sequence with $N = 10000$ as benchmark. The $N$ trajectories of the system in Eq. (2) are computed with a Runge-Kutta scheme of order 4 (RK4).

The empirical measure of the snapshot attractors is estimated at each time step $t$ with the algorithm 2. We then compute the Wasserstein distance between those time varying snapshot attractors, and four reference seasonal snapshot attractors obtained for $t = 0$, 18.25, 36.5 and 54.75, during the first year. The four reference seasonal snapshot attractors are shown in Figure 5(a-d), with $N = 10000$ points. For illustration purposes, the snapshot attractors corresponding to the same seasons, but at year 180, after the monotonous forcing is triggered. It is obvious from Figure 5(e-f) that the forcing affects each of the seasonal

attractors.

The yearly averages of the distances to the four reference attractors are shown in Figure 6. We detect the change point, with a trend, after $t = 100$ years. Therefore, the detection of the forcing effect on the dynamics of the Lorenz84 system is rather immediate, with a lag $< 10$ years.

We find that the variability of the distance variations decrease with the number $N$ of points to estimate the snapshot attractors, although it does not seem to affect the detection of the change point. Relatively low values of $N$ show a bias of the distance, which is even higher for lower values of $N$. The mean values of the attractor distance distributions is quite similar if $N \geq 100$. This sets a lower bound for the number of points to estimate the measure of snapshot attractors.

In this example, the distances of the snapshot attractors to winter and Spring reference attractors increase with time after $t = 100$ years. Conversely, the distance to Fall and summer reference attractors decrease with time. We interpret this as a shift of all snapshot attractors toward "hot" conditions.

Those results are consistent with those of Drótos et al. (2015). The main practical value of our approach is that the number of points that is needed to sample snapshot attractors can be as low as $N = 100$, rather than $N = 10^6$, which is generally not available.

The same experiment is conducted with the Euclidean distance. For $N = 50$ and $100$, the maximal difference of the mean (resp. standard deviation) between the period before and after the forcing is $0.002$ (resp. $0.002$), whereas the mean is $0.2$ (resp. $0.002$) . Furthermore, at least $70\%$ of distances are in the pip of mean plus or minus standard deviation. So, we can not detect the forcing. For $N = 1000$ and $10000$ the mean is $0.08$, and its the maximal modification is $0.004$. The standard deviation is multiplied by a factor $20$ ($0.0002$ becomes $0.005$). Even if the forcing is detected, the trajectories of distances are not representative of a linear increasing forcing.

## 5 Conclusions

The Wasserstein distance appears to be efficient to measure changes in the dynamics in time evolving systems even with a relatively low number of points (e.g. $N = 100$). This discrimination is still powerful when only partial information on the attractor is available (truncation and/or projection). We made the assumption that the system we investigate yields an attractor, and the Wasserstein distance determines changes in the invariant measure of the attractors. This builds an interesting bridge between dynamical systems and optimal transport. A theoretical justification for this bridge is recalled in Appendix B. A caveat of the approach we present here is that we do not give an interpretation of the Wasserstein distance in terms of qualitative dynamical changes (e.g. changes in local dimensions (Faranda D. et al., 2017)). Villani (2003, Chapter 9) provides links between the Wasserstein distance and entropy, but they are hard to interpret and infer for the problem we tried to tackle (measure a change in a strange attractor).

The main caveat of this approach is its computational cost. The minimization of the cost function, constraint by the estimated measures, has to be implemented by network simplex algorithms (Bazaraa et al., 2009; Boyd and Vandenberghe, 2004; Dantzig et al., 1955; Gottschlich and Dominic, 2014). Those algorithms are computationally expensive, but applicable, as shown with the Lorenz 84 model ($200,000$ distances computed in sixty hour on 12 cores).

A research challenge would be to adapt this method on climate model simulations from CMIP5 (Taylor et al., 2012). The Wasserstein distance could be computed to discriminate between control, natural and historical runs.

## Appendix A: Computation of the Wasserstein distance

We just give here the general idea to compute Wasserstein distance with the network simplex algorithm. We want to transport the measure $\mu$ to $\nu$, can be written

$$\mu = \sum_{i=1}^{n} \mu_i \delta_{\mathbf{x}_i}, \quad \nu = \sum_{j=1}^{p} \nu_j \delta_{\mathbf{y}_j}.$$

The Wasserstein distance is given by minimizing over $\gamma_{ij}$ (the mass transported from $\mathbf{x}_i$ to $\mathbf{y}_j$) the cost function

$$\sum_{ij} \gamma_{ij} \mathbf{d}(\mathbf{x}_i, \mathbf{y}_j)^2.$$

Consequently, we have the following linear constraints:

$$\mu_i = \sum_{j=1}^{p} \gamma_{ij}, \ \nu_j = \sum_{i=1}^{n} \gamma_{ij}, \ \gamma_{ij} \geq 0$$

These constraints define a polyhedral convex set in the space of $\gamma_{ij}$. The solutions of all constraints are the extremal point of the polyhedra, and the $\mathcal{W}$ distance is one of its minima. The network simplex algorithm runs in two part:

1. Finding a first extremal point.

2. Iterate over the face of polyhedra (the simplex) until the minimal solution is reached.

Because the number of extremal point increases exponentially with the size of data, this algorithm has an exponential complexity. But, in practice the iteration over simplex are made in the direction of an optimal solution. Thus, it has been found that the complexity of the algorithm is polynomial in practice. Currently, we use a C++ implementation of the R-package *transport* (Baehre et al., 2016), using the methodology described in Gottschlich and Dominic (2014). We have also tested entropy regularization (Cuturi, M., 2013). This algorithm cross the polyhedra until the optimal solution, but it requires a parameter changing for each distance. We preferred to use the network simplex method, which work all time.

## Appendix B: Theoretical justification

Besides the simulations studied in the previous sections, it is possible to theoretically justify the use of the Wasserstein distance for nonautonomous dynamical systems. Any dynamical system defined from an ordinary differential equation, say $\frac{\mathrm{d}\mathbf{x}}{\mathrm{d}t} = \mathbf{v}(\mathbf{x}, t)$, is formally equivalent (e.g., see Villani, 2003; Evans, 2010) to the partial differential equation of a transport of the density of trajectories of the associated dynamical system, say $\frac{\partial \rho_t}{\partial t} + \langle \nabla, \rho_t \mathbf{v} \rangle = 0$. In other words, the variations between $t_0$ and $t_1$ of the time-varying attractor in $\frac{\mathrm{d}\mathbf{x}}{\mathrm{d}t} = \mathbf{v}(\mathbf{x}, t)$ can be determined by the transport of the measure of the attractors by the dynamics $\mathbf{v}$. If $\mu_t$ denotes the density distribution of $\rho_t$ (i.e. $\mu_t(A) = \int_A \rho_t(\mathbf{x}).\mathrm{d}\mathbf{x}$), then the Wasserstein distance between $\mu_{t_0}$ and $\mu_{t_1}$ for attractors in dimension $d$ is given by the so-called Benamou-Brenier theorem (Benamou and Brenier, 1998)

$$\mathcal{W}(\mu_{t_0}, \mu_{t_1})^2 = \frac{1}{t_1 - t_0} \inf_{(\tilde{\rho}_t, \tilde{\mathbf{v}})} \int_{\mathbb{R}^d} \int_{t_0}^{t_1} \tilde{\rho}_t(\mathbf{x}) |\tilde{\mathbf{v}}(\mathbf{x}, t)|^2 \, dt \, d\mathbf{x}.$$

The minimization is done over all vector fields $\tilde{\mathbf{v}}$ and all sequences of density $\tilde{\rho}_t$ following $\tilde{\mathbf{v}}$ such that $\tilde{\rho}_{t_0} = \rho_{t_0}$ and $\tilde{\rho}_{t_1} = \rho_{t_1}$. This theorem connects the dynamical systems theory with the optimal transport theory. Therefore, the Wasserstein distance between two snapshot attractors of a time varying dynamical system is linked to the energy ($\mathbf{v}$ is homogeneous to a velocity) of the system that transforms one attractor into the other. If the dynamics $\mathbf{v}$ is unknown and only simulations are available, this theorem allows (in principle) to infer $\mathbf{v}$ from the simulations because the optimum path going from the snapshot attractor at $t_0$ to $t_1$ is achieved by the actual dynamics $\mathbf{v}$.

5  *Author contributions.* Y.R. conducted the computations and produced the figures. The idea of the experiments were formulated by the three authors. The three authors contributed to writing the manuscript.

*Competing interests.* The authors declare no conflict of interest.

*Acknowledgements.* This work is supported by ERC grant No. 338965–A2C2. We thank Davide Faranda and Ara Arakelian for useful discussions.

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

**Table 1.** Kolmogorov-Smirnov test applied between distribution $\mathcal{W}(\mu^{\mathrm{Wi}}, \mu^{\mathrm{Wi}})$ (resp. $\mathcal{W}(\mu^{\mathrm{Su}}, \mu^{\mathrm{Su}})$) and $\mathcal{W}(\mu^{\mathrm{Wi}}, \mu^{\mathrm{Su}})$ in left (resp. right) box on snapshots of size $N$.

| $N$ | | 50 | | 100 | | 1000 | |
|---|---|---|---|---|---|---|---|
| Wasserstein distance | KS-test | 0.98 | 0.99 | 1 | 1 | 1 | 1 |
| | $p$-value | 0 | 0 | 0 | 0 | 0 | 0 |
| Euclidean distance | KS-test | 0.71 | 0.97 | 0.77 | 1 | 0.47 | 1 |
| | $p$-value | 0 | 0 | 0 | 0 | 0 | 0 |

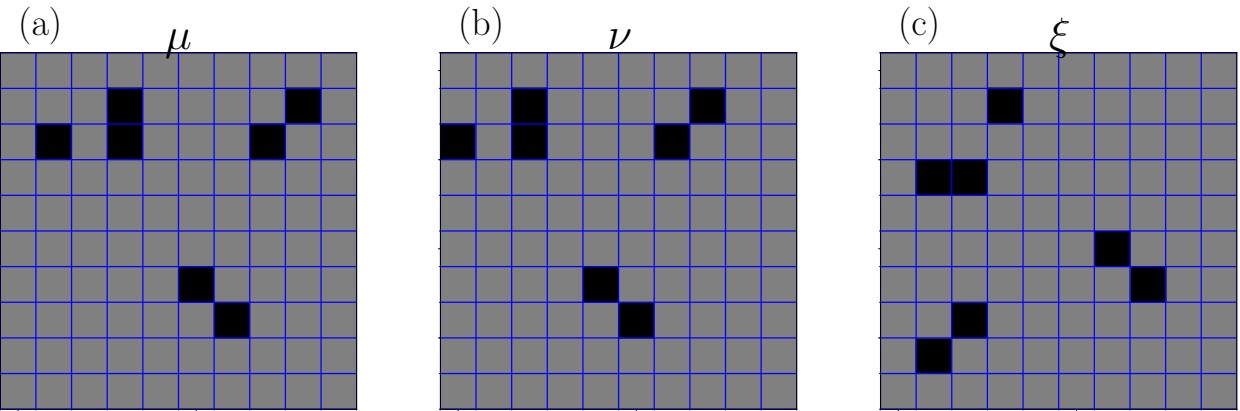

**Figure 1.** $\mu$, $\nu$ and $\xi$ are three examples of density of attractors. The black boxes have a measure of $1$ and the grey a measure of $0$. $\nu$ is a shift of $\mu$, but $\xi$ is very different of $\mu$ and $\nu$. Finally, $\nu$ (resp. $\xi$) have no common (resp. one common) point with $\mu$. The Euclidean distance between $\mu$ and $\nu$ (resp. $\xi$) is equal at $3.74$ (resp. $3.46$), whereas the Wasserstein distance is equal to $1$ between $\mu$ and $\nu$, and $3.27$ between $\mu$ and $\xi$.

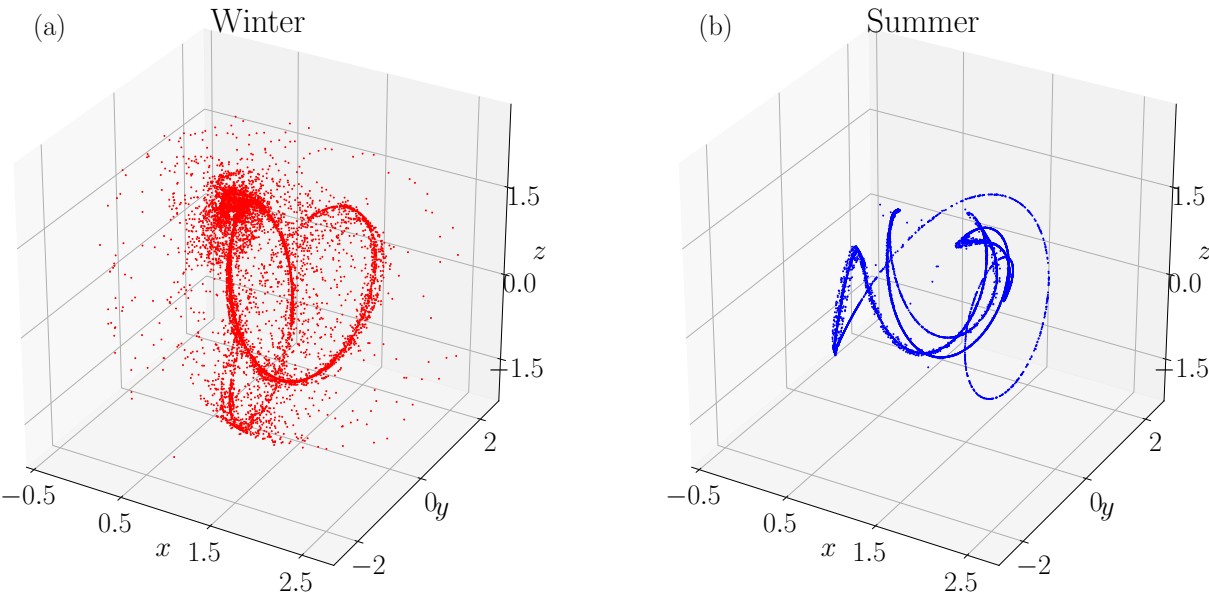

**Figure 2.** (a) winter snapshot attractor of the Lorenz84 model. (b) summer snapshot attractor of the Lorenz 84 model. Each of $10,000$ points is the solution at time $5\tau = 5 \times 73$ of the Lorenz 84 equation (see Eq. (2)), and constructed with a time step of integration of $0.005$ using RK4 scheme.

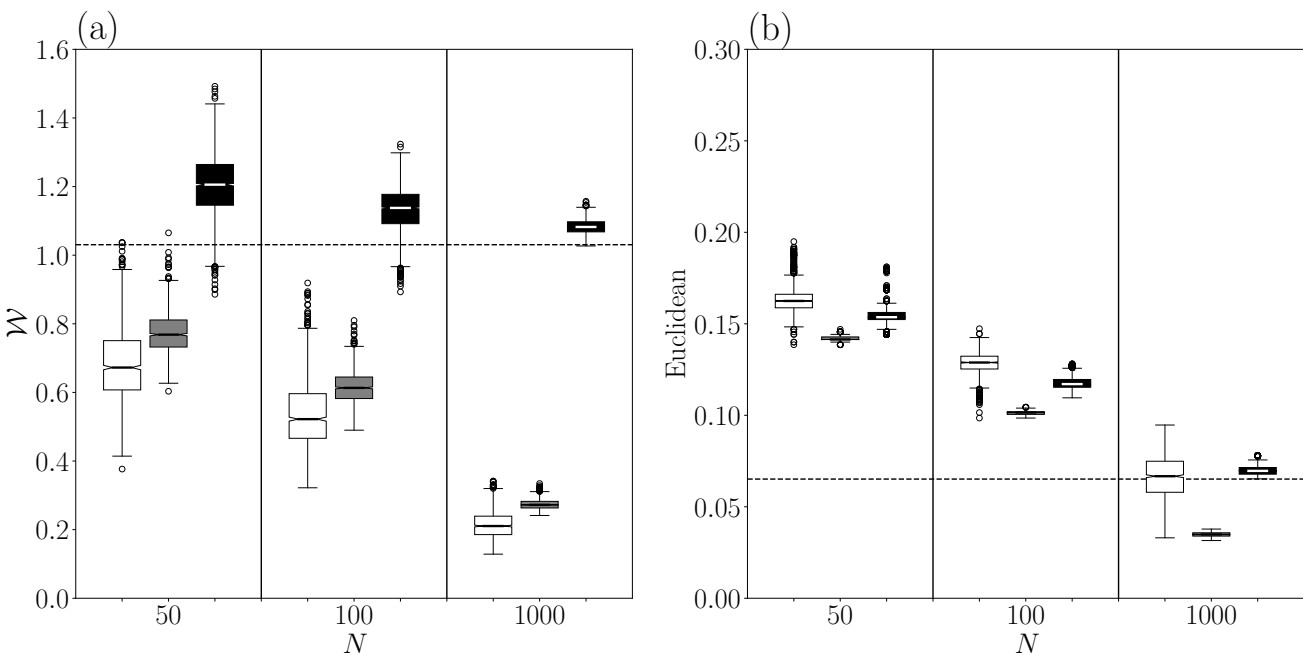

**Figure 3.** Boxplots of distances computed using the Wassertein distance (left panel) and the Euclidean one (right panel). White boxplots differentiate between two winter snapshots. Grey boxplots differentiates two summer snapshots. Black boxplots compare winter and summer snapshots. Dotted lines represent the distance between winter and summer attractors with $N = 10^6$ points.

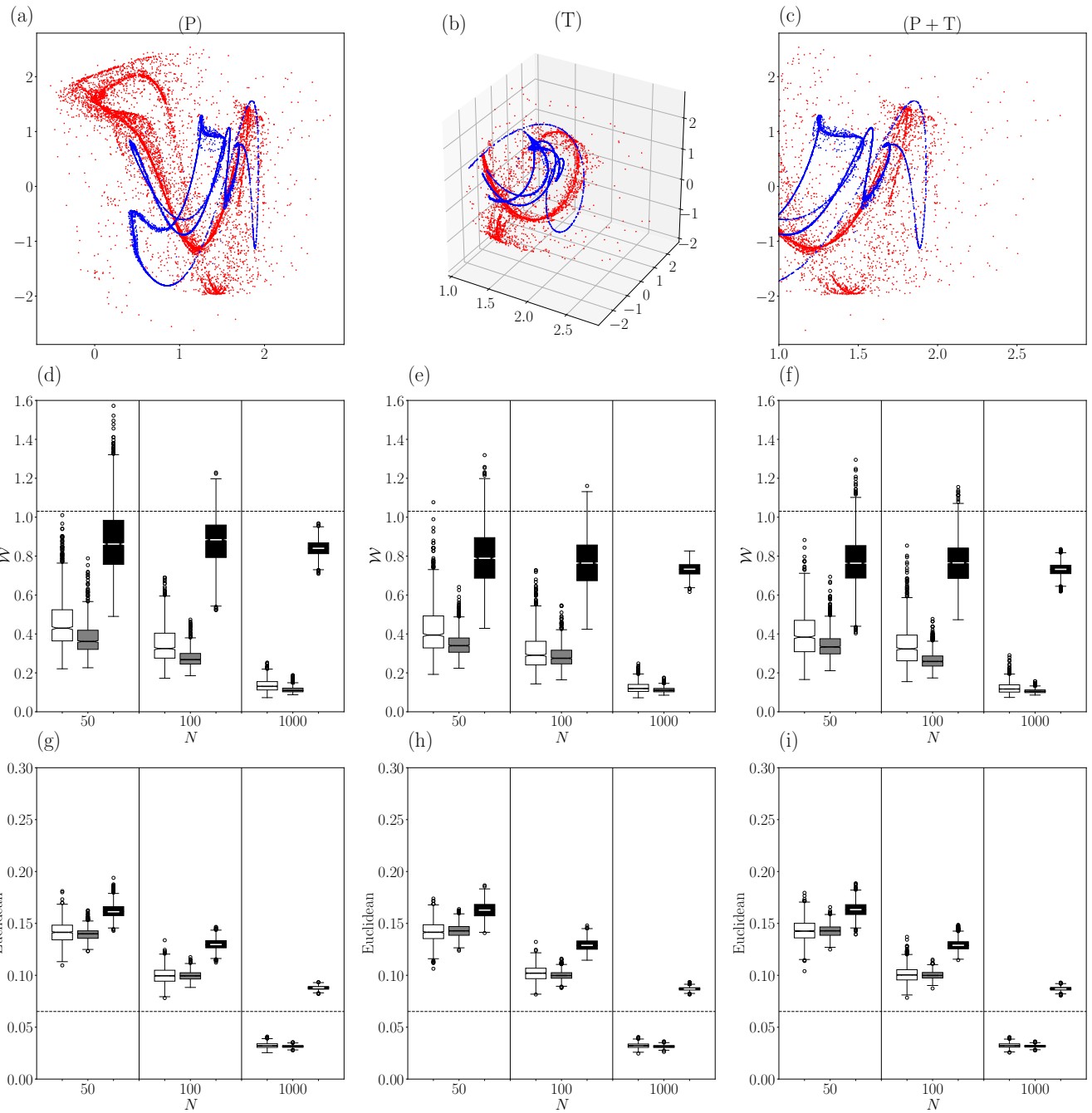

**Figure 4.** (a) Projection on axes $(x, y)$ of the winter snapshot (red) and summer snapshot (blue). (b) Truncature at $x_1 \geq 1$ of the winter and summer. (c) Combination of (a) and (b) (see designs (T), (P) and (T+P) in Section 3.4). (d-f) Boxplots of distances between design T, P and T+P computed with the Wassertein distance. (g-i) Boxplots of distances between design T, P and T+P computed with the Euclidean distance. (d-i) White boxplots differentiate between two winter snapshots. Grey boxplots differentiates two summer snapshots. Black boxplots compare winter and summer snapshots. Dotted lines represent the distance between winter and summer attractors (without designs) with $N = 10^6$ points.

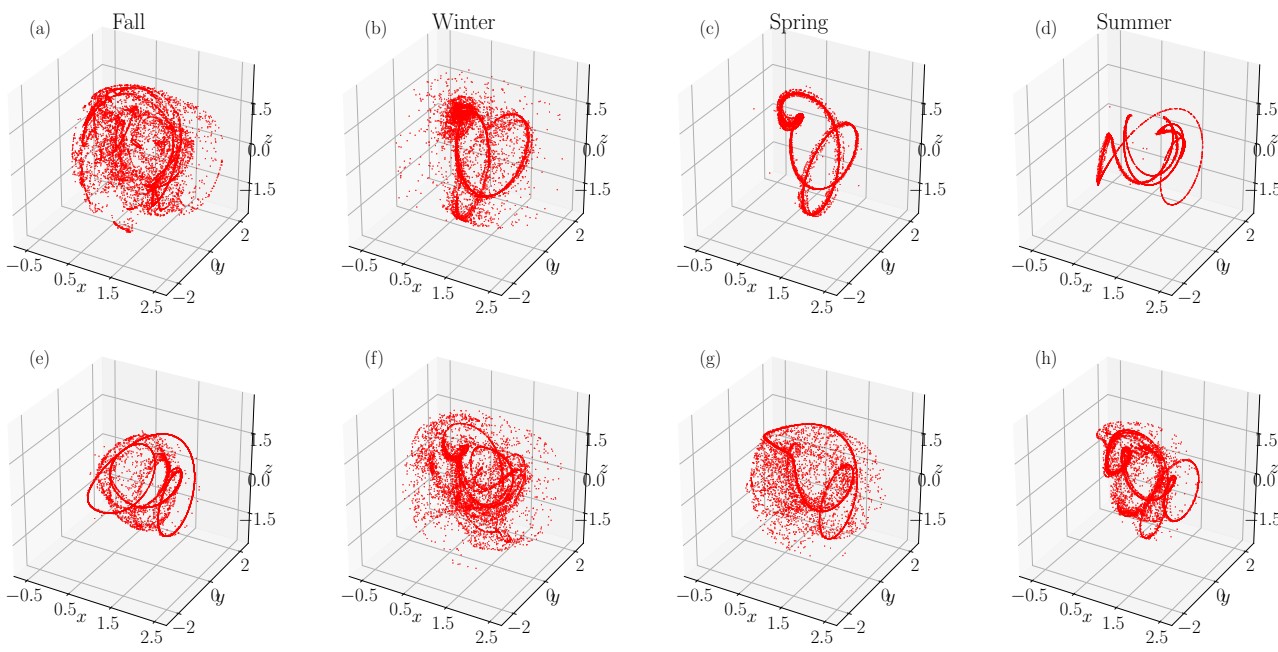

**Figure 5.** Snapshots of 10.000 points from the Lorenz 84 defined by Eq. (2). (a-d) The four seasons at time $t = 0$ year, $t = 0.25$y., $t = 0.5$y. and $t = 0.75$y. (e-h) The same seasons, but after the triggering of the linear forcing, during year 180.

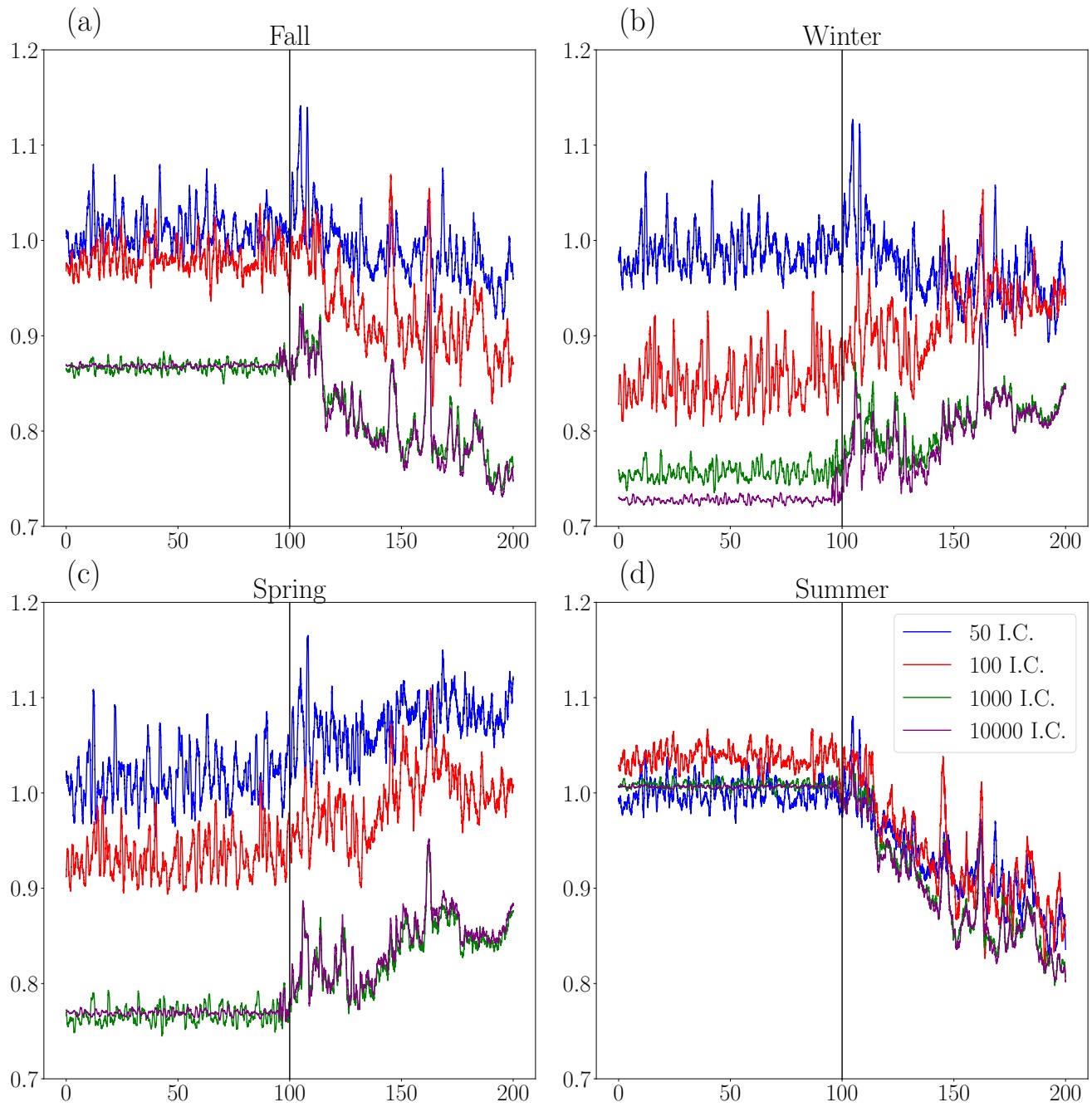

**Figure 6.** Yearly averages Wasserstein distance between the reference attractor before forcing, and all other attractors. The $x$ axis is the time, the $y$ axis the estimated Wasserstein distance. The blue (resp. red, green and purple) is the numbers of initial conditions (I.C.) for $N = 50$ (resp. 100, 1000 and 10.000). The vertical black line represents the instant when the linear trend is triggered in the forcing $F(t)$.