# Peer review of "Detecting Changes in Forced Climate Attractors with Wasserstein Distance"

_Nonlinear Processes in Geophysics, 2017_

## Referee Comment (RC1) · Anonymous Referee #1 · 8 Mar 2017

The authors propose the use of the Wasserstein distance in order to discriminate different dynamical systems from their attractors, notably for the case of climate systems. I found the paper really interesting. Moreover, the adoption of such new metric is well motivated and seems really promising for future climatic applications. Thus, I recommend the publication of the manuscript. I have only a general comment and a few specific ones (see below) that could be useful to improve the manuscript.

GENERAL COMMENT:

Did the authors studied the robustness of the Wasserstein distance to variations of the size of the boxes $B_{a}$? I think this is an important point, in particular once that their method will be applied to realistic systems. Even though an explicit numerical study is not requested, a discussion of this aspect would be really appreciated.

SPECIFIC COMMENTS:

(2 Distance between measures - line 14) : It could be not easy for any reader how you go from attractors to mass distributions. It would be great to have a short introduction to the definition and use of invariant measures in phase space.

(2 Distance between measures - line 22) : Why the authors did not defined (and discuss the differences respect) the Mahalanobis distance?

(2 Distance between measures - line 11 - second paragraph) : It would be interesting to know why the authors choose network simplex algorithms to compute the distance. Could be explained why they are better than other classical choices like, for instance, simulated annealing algorithms?

(Algorithm 2) : Maybe the authors could give a name to the variable: "total number of boxes $B_{a}$" like, for example, K.

(3.2 Protocol line – line 16,17) : This sentence is not really clear. It could be expanded a bit.

(3.3 Estimation – line 1) : The first sentence is not really clear.

(3.3 Estimation – line 11 to 15) : This point is interesting and could be linked to my general comment: which is the sensitivity of the method respect to N together with the number of boxes $B_{a}$? Probably such parameters present an interplay in determining the global robustness of the measure.

(3.3 Estimation – line 4 – second paragraph) : Could the authors specify how they computed the p-values for the KS test? Did they use tables of critical values or simulated numerical p-values?

(4.1 Protocol line – line 31) : Could the authors show also here the comparison with the Euclidean distance? Why they did not show such calculation?

---

## Referee Comment (RC2) · V. Lucarini (Referee) · 6 May 2017

Dear authors,

I have found the paper quite interesting and innovative and I support its publication in NPG once certain issues are analysed in greater detail. I would like to make some remarks that I hope the authors will take into consideration.

1- Page 1, Line 21

The authors should consider giving a look Lucarini et al. J Stat Phys 166 1036–1064 (2017) where an extensive statistical mechanical analysis of climate response to forcing is given.

2 - Page 2, Line 24. "Intuitive" is not really a good world. Our visual impression and

the way we interpret it is far from being in any sense objective. I understand what the authors say, but I kindly ask to re-formulate.

3 - Page 4, line 18. The construction of the pullback attractor requires the integrations started at a t=t_0, with t_0 going to minus \infty. Otherwise no well-posed definition is possible. This should be clearly explained. Is one year of integration enough, in this case?

4 - Page 5, line 11. In this part there is no mention of the way A is chosen. This seems quite important for the rest of the paper.

5 - Page 5, line 13. The authors might want to note explicitly that each of the realised estimate of the measure supported by the pullback attractor come from initial conditions at t_0 (see point 3) distributed uniformly according to Lebesgue of the union of the little cubes.

6 - Page 6, Section 3.3 Discussion on the value of A is missing.

7 - Page 7, line 3 - I disagree with the use of "visual impression".

8 - Page 8, lines 13-14 - The statement is indeed overblown if given in all generality as here.

9 - Page 9 - End of Section 4. There is a fundamental misunderstanding here, I believe. It is true that a much lower number of integrations is needed to say that two attractors are different. This is a very interesting result. But you are not able to quantify well what is (quantitatively) the difference between the expectation value of any given (possibly interesting) observable of relevance. So, you are left with a statement that is in fact qualitative rather than quantitative (the two attractors are different!). How can you relate the Wasserstein measure to any useful information?

This does NOT diminish the relevance of the performed analysis, to be clear.

10 - Page 11, line 7: not clear the relationship between \rho and \mu.

Best Regards, Valerio Lucarini
* * *
Interactive
comment

---

## Author Comment (AC2) · 8 Jun 2017

*I have found the paper quite interesting and innovative and I support its publication in NPG once certain issues are analysed in greater detail. I would like to make some remarks that I hope the authors will take into consideration.*

**(1 - Page 1, Line 21)**: The authors should consider giving a look Lucarini V. (2017) where an extensive statistical mechanical analysis of climate response to forcing is given.

**Response**: The reference has been added.

**(2 - Page 2, Line 24)**: "Intuitive" is not really a good world. Our visual impression and

the way we interpret it is far from being in any sense objective. I understand what the authors say, but I kindly ask to re-formulate.

**Response**: We agree with you.

**Modification** (Page 2, line 16): "Although it is intuitive from Figure 1 that..." has been replaced by "Panels a and b in Figure 1 are visually very similar, whereas Panel c cannot be deduced from a trivial transformation of the first panels. Therefore, it is expected that $\mu$ is "closer" to $\nu$ than $\xi$.".

**(3 - Page 4, line 18)**: The construction of the pullback attractor requires the integrations started at a $t = t_0$, with $t_0$ going to minus $\infty$. Otherwise no well-posed definition is possible. This should be clearly explained. Is one year of integration enough, in this case?

**Response**: In Section 3, the integration is performed between $0$ and $\tau$. Because the dynamic of Lorenz 84 for winter and summer does not depend on time, it is equivalent to an integration between $-\tau$ and $0$. The parameter $\tau$ is chosen to be $5$ "years", but we checked than $1$ year is enough. Following Drotos et al (2015), we keep $\tau$ at $5$ years. In Section 4, a first integration is performed between $0$ and $\tau$, and then we integrate between $0$ and $200 \times 73$ from first integration. Due to cyclicity of seasonal forcing, the first integration is equivalent to an integration between $-\tau$ and $0$.

**Modification** (Page 4, line 16): The parameter $\tau$ has been set at one year ($\tau = 73$), and the first integration is performed during $5\tau$ (i.e. $5$ cycles / years). We have added the sentence "Snapshot attractors are special cases of *pullback attractors* Chekroun et al (2011). The latter class requires an integration between $-\infty$ and a desired final time. Eq. (2) does not depend of time, so the integration into Sec. 3.1 can be performed on any length intervals".

**(4 - Page 5, line 11)**: In this part there is no mention of the way $A$ is chosen. This seems quite important for the rest of the paper.

**Response**: See Question 6.

**(5 - Page 4, line 13)**: The authors might want to note explicitly that each of the realised estimate of the measure supported by the pullback attractor come from initial conditions at $t_0$ (see point 3) distributed uniformly according to Lebesgue of the union of the little cubes.

**Response**: We agree with you

**Modification** (Page 4, line 11): We have replaced "we draw $N$ random initial conditions in a cube that includes the attractors, and iterate the dynamics of the systems for a time $\tau$", by "we draw $N$ random initial conditions following a uniform distribution. All margins are independent. This approximates a Lebesgue measure in a cube that includes the attractors. We iterate the dynamics of the systems between $t_0 = 0$ and a long time multiple of $\tau$."

**(6 - Page 6, Section 3.3)**: Discussion on the value of $A$ is missing

**Response**: $A$ was a misleading parameter. It represents only the number of boxes with non zero mass, but it can be different for different realization of same attractor. The value of $0.1$ gives $40$ to $60$ bins on each axis, assuming that the attractor lives in a box of $[-1; 3] \times [-3; 3] \times [-3; 3]$. This means that the volume is divided into $40 \times 60 \times 60$ boxes. This number is the same order of magnitude as the number of gridcells in the NCEP reanalysis around the North Atlantic region, should one be interested in the climate attractor of that region (e.g. Faranda D. et al (2017)). We also tried values of $0.05$, $0.2$ and $1.0$ for the size of the boxes (so, a factor $2$ for the two first, and one scale up for the last) for the protocol of Section 3. For all values, the maximal variation

Interactive
comment

of standard deviation is $0.01$, and the detection is not affected. For a size of $0.05$ and $0.2$ the maximal variation of median is $0.03$. For the size $1.0$, the maximal increases of median of box plot of winter (resp. summer) against itself is $0.22$ (resp. $0.18$), but the difference with median of winter against summer is at least equal to $0.3$.

**Modification 1** (Page 5, line 5): We have been added the sentence: "We chose a bin length of $0.1$ for the Lorenz attractor, which remains in a $[-1; 3] \times [-3; 3] \times [-3; 3]$ box. Therefore $40 \times 60 \times 60$ bins cover the attractor. This number of bins is comparable to the number of gridcells that cover the North Atlantic region in the NCEP reanalysis (or most CMIP5 model simulations). This example refers to a few papers dealing with climate attractor properties (e.g. Corti S. et al (1999) and Faranda D. et al (2017)).

**Modification 2** (Page 7, line 15): The sentence "This protocol was also applied for bin sizes of $0.05$, $0.2$ and $1.0$. For $0.05$ and $0.2$, the maximal variation of median (resp. standard deviation) of Wasserstein distances is $0.03$ (resp. $0.01$), so the distributions are indistinguishable in practice. For a bin size of $1.0$, the maximal increase of the median is $0.22$, but the difference with the median of winter against summer is at least equal to $0.3$." has been added at the end of Sec. 3.3.

**(7 - Page 7, line 3)**: I disagree with the use of "visual impression".

**Response**: We agree with you.

**Modification** (page 7, line 11): "This visual impression is confirmed..." has been replaced by "This discrimination is confirmed..."

**(8 - Page 8, lines 13-14)**: The statement is indeed overblown if given in all generality as here.

**Response**: We agree with you.

**Modification** (page 8, line 25): "We conclude that the Wasserstein distance has a high

capacity of discriminating between different dynamical systems" has been replaced by "We conclude that the Wasserstein distance has a high capacity of discriminating different attractors coming from this dynamical system"

**(9 - Page 9 - End of Section 4)**: There is a fundamental misunderstanding here, I believe. It is true that a much lower number of integrations is needed to say that two attractors are different. This is a very interesting result. But you are not able to quantify well what is (quantitatively) the difference between the expectation value of any given (possibly interesting) observable of relevance. So, you are left with a statement that is in fact qualitative rather than quantitative (the two attractors are different!). How can you relate the Wasserstein measure to any useful information? This does NOT diminish the relevance of the performed analysis, to be clear.

**Response**: It is true we do not use the link between Wasserstein distance and other dynamical information to measure qualitative changes in attractors (e.g. dimensions). The chapter 9 of Villani (2003) gives some link between the Wasserstein distance and entropy.

**(10 - Page 11, line 7)**: not clear the relationship between $\rho$ and $\mu$.

**Response**: $\rho$ is the density of the measure $\mu$.

**Modification** (page 11, line 25): We state that: $\rho(\geq 0)$ is the density of the measure $\mu$. Hence, for all Borel set $A$ in phase space, they are related by:

$$\mu_t(A) = \int_A \rho_t(\mathbf{x}).\mathrm{d}\mathbf{x}.$$

**References**

Chekroun, M. D., Simonnet, E., and Ghil, M.: Stochastic climate dynamics: Random attractors and time-dependent invariant measures, Physica D, 240, 1685–1700,

doi:10.1016/j.physd.2011.06.005, 2011.

Corti, S., Molteni, F., and Palmer, T. N.: Signature of recent climate change in frequencies of natural atmospheric circulation regimes, Nature, 398, 799–802, doi:10.1038/19745, 10.1038/19745, 1999.

Drótos, G., Bódai, T., and Tél, T.: Probabilistic concepts in a changing climate: a snapshot attractor picture, J. Climate, 28, 3275–3288, doi:10.1175/JCLI-D-14-00459.1, http://dx.doi.org/10.1175/JCLI-D-14-00459.1, 2015.

Faranda, D., Messori, G., and Yiou, P.: Dynamical proxies of North Atlantic predictability and extremes, Sci. Rep., 7, doi:10.1038/srep41278, http://dx.doi.org/10.1038/srep41278, 2017.

Lucarini, V., Ragone, F., and Lunkeit, F.: Predicting Climate Change Using Response Theory: Global Averages and Spatial Patterns, J. Stat. Phys., 166, 1036–1064, doi:10.1007/s10955-016-1506-z, http://dx.doi.org/10.1007/s10955-016-1506-z, 2017.

Villani, C.: Topics in Optimal Transportation, 58, American Mathematical Society, 2003.

---

## Author Response (AR1)

Dear Editor,

Please find in this document:
- The reply to referee #1 (page 2 to 8)
- The reply to referee #2 (page 9 to 14)
- The revised manuscript with track change in blue (starting at page 15).

Sincerely,

Y. Robin, P. Yiou, P. Naveau

**Response Referee # 1**

Yoann Robin[1], Pascal Yiou[1], and Philippe Naveau[1]

[1]LSCE, Gif-sur-Yvette, France

*Correspondence to:* Y. Robin (yoann.robin@lsce.ipsl.fr)

*The authors propose the use of the Wasserstein distance in order to discriminate different dynamical systems from their attractors, notably for the case of climate systems. I found the paper really interesting. Moreover, the adoption of such new metric is well motivated and seems really promising for future climatic applications. Thus, I recommend the publication of the manuscript. I have only a general comment and a few specific ones (see below) that could be useful to improve the manuscript.*

5 **General comment**

*Did the authors studied the robustness of the Wasserstein distance to variations of the size of the boxes $B_a$? I think this is an important point, in particular once that their method will be applied to realistic systems. Even though an explicit numerical study is not requested, a discussion of this aspect would be really appreciated.*

10 **Response**

The value of $0.1$ gives $40$ to $60$ bins on each axis, assuming that the attractor lives in a box of $[-1;3] \times [-3;3] \times [-3;3]$. This means that the volume is divided into $40 \times 60 \times 60$ boxes. This number is the same order of magnitude as the number of gridcells in the NCEP reanalysis around the North Atlantic region, should one be interested in the climate attractor of that region (e.g. Faranda D. et al. (2017)). We also tried values of $0.05$, $0.2$ and $1.0$ for the size of the boxes (so, a factor 2 for the
15 two first, and one scale up for the last) for the protocol of Section 3. For all values, the maximal variation of standard deviation is $0.01$, and the detection is not affected. For a size of $0.05$ and $0.2$ the maximal variation of the median is $0.03$. For the size $1.0$, the maximal increases of median of box plot of winter (resp. summer) against itself is $0.22$ (resp. $0.18$), but the difference with the median of winter against summer is at least equal to $0.3$.

**Modification (Page 5, line 5 and page 7, line 15)**

20 We have added the sentence (end of section 3.1): "We chose a bin length of $0.1$ for the Lorenz attractor. Therefore $40 \times 60 \times 60$ bins cover the attractor, which remains in a $[-1;3] \times [-3;3] \times [-3;3]$ box. This number of bins is comparable to the number of gridcells that cover the North Atlantic region in the NCEP reanalysis (or most CMIP5 model simulations). This example refers to a few papers dealing with climate attractor properties (e.g. Corti S. et al. (1999); Faranda D. et al. (2017)).

The sentence "This protocol was also applied for bin sizes of 0.05, 0.2 and 1.0. For 0.05 and 0.2, the maximal variation of median (resp. standard deviation) of Wasserstein distances is 0.03 (resp. 0.01), so the distributions are indistinguishable in practice. For a bin size of 1.0, the maximal increase of the median is 0.22, but the difference with the median of winter against summer is at least equal to 0.3." has been added at the end of Sec. 3.3.

**5 Specific comments**

**1 2 Distance between measures - line 14**

It could be not easy for any reader how you go from attractors to mass distributions. It would be great to have a short introduction to the definition and use of invariant measures in phase space.

**Response**

We agree with you.

**Modification (Page 3, line 23)**

We added the sentence "The measure of a sub region of phase space is the probability of a trajectory of the system to go through the region. The invariance is characterized by the conservation of the volume by the dynamics of the system (Ruelle, 1989)" in Sec. 3.1.

**2 2 Distance between measures - line 22**

*Why the authors did not defined (and discuss the differences respect) the Mahalanobis distance?*

**Response**

The Mahalanobis distance was just given as an example of possible distances used in climate sciences. We removed the reference to the Mahalanobis distance, since we do not make any comparison with it.

**3 2 Distance between measures - line 11 - second paragraph**

*It would be interesting to know why the authors choose network simplex algorithms to compute the distance. Could be explained why they are better than other classical choices like, for instance, simulated annealing algorithms?*

**Response**

The optimal transport literature classically mentions two kinds of methods: Network Simplex and Entropy Regularization. These two approaches have the advantage to be computationally fast. The Network Simplex is generic but the Entropy Regularization needs a control parameter to be adapted for each system. Thus we have decided to use the Network Simplex for this paper. Annealing algorithms require also to test several control parameters (like acceptance probabilities and temperature) depending on the measures considered. This could be problematic for the computation of thousands of distances between various objects.

**5 Modification (Appendix A)**

An explanation has been added in Appendix A.

**4 Algorithm 2**

*Maybe the authors could give a name to the variable: "total number of boxes $B_a$" like, for example, $K$.*

**Response**

The variable "$A$" (in Require/ Ensure of Algorithm 2) is the total number of boxes.

**Modification (Algorithm 2)**

To clarify, we have replaced $A$ by $40 \times 60 \times 60$ (the total number of boxes of size $0.1$ in the domain $[-1;3] \times [-3;3] \times [-3;3]$) and explained that $\mu_a > 0$ for a small number boxes.

**5 3.2 Protocol line – line 16,17**

*This sentence is not really clear. It could be expanded a bit.*

**Response**

We agree with you.

**Modification (Page 6, line 5)**

"We choose to simulate $50$ attractors of winter and $50$ attractors of summer. We have $50 \times 50 = 2500$ different pairs between summer and winter. For the distances between the $50$ attractors of the same season (summers or winters), we only consider $1 \leq (k, k') \leq 50$ pairs with $k < k'$. This means that we have $1225$ distances for the winter or the summer. So we have at least $1000$ distances per distribution. This is a reasonable sample size for a representative Kolmogorov-Smirnov test."

**6 3.3 Estimation – line 1**

*The first sentence is not really clear.*

**Modification**

Normalize Wasserstein distance do not add information in our protocol, so the sentence has been removed.

**7  3.3 Estimation – line 11 to 15**

*This point is interesting and could be linked to my general comment: which is the sensitivity of the method respect to $N$ together with the number of boxes $B_a$? Probably such parameters present an interplay in determining the global robustness of the measure.*

**Response**

See general comment for the question. The global robustness of the empirical measure could be estimated by varying this parameter.

**8  3.3 Estimation – line 4 – second paragraph**

*Could the authors specify how they computed the p-values for the KS test? Did they use tables of critical values or simulated numerical p-values?*

**Response**

The KS value is computed with an estimation of the cumulated density function of the two distributions, and the difference. The $p$-value is given by the asymptotic Kolmogorov distribution. Its cumulative distribution function converges to the supremum of a Brownian bridge $B$, which can be computed with

$$\mathbb{P}(K \leq x) = 1 - 2\sum_{k=1}^{\infty}(-1)^{k-1}\mathrm{e}^{-2k^2x^2}, \;\; K = \sup_{t\in[0,1]}|B(t)|$$

This formula can be found in (Marsaglia G. et al., 2003)

**Modification (Page 6, line 3)**

Reference and explanation have been added.

**9  4.1 Protocol line – line 31**

*Could the authors show also here the comparison with the Euclidean distance? Why they did not show such calculation?*

**Response**

We show in Figure 1 below the calculation for Euclidean distance. For $N = 50$ and 100, the maximal difference of the mean (resp. standard deviation) between the period before and after the forcing is $0.002$ (resp. $0.002$). Furthermore, at least $70\%$ of distances are in the pip of mean $\pm$ standard deviation. So, we can not detect the forcing. For $N = 1000$ and 10000 the maximal modification of mean is $0.004$, but the standard deviation is multiplied by a factor 20 ($0.0002$ becomes $0.005$). Even if the forcing is detected, the trajectories of distances are not representative of a linear increasing forcing. This calculation was not added in article because we find the same result of Section 3, and we focus only on the Wasserstein distance.

**Modification (Page 10, line 11)**

This explanation has been added at the end of Sec. 4.

[Figure]

**Figure 1.** Yearly averages Euclidean distance between the reference attractor before forcing, and all other attractors. The $x$ axis is the time, the $y$ axis the estimated Euclidean distance. The blue (resp. red, green and purple) is the numbers of initial conditions (I.C.) for $N = 50$ (resp. 100, 1000 and 10.000).

**Response Referee # 2**

Yoann Robin[1], Pascal Yiou[1], and Philippe Naveau[1]

[1]LSCE, Gif-sur-Yvette, France

*Correspondence to:* Y. Robin (yoann.robin@lsce.ipsl.fr)

*I have found the paper quite interesting and innovative and I support its publication in NPG once certain issues are analysed in greater detail. I would like to make some remarks that I hope the authors will take into consideration.*

**1 Page 1, Line 21**

*The authors should consider giving a look Lucarini, V. et al. (2017) where an extensive statistical mechanical analysis of*
5 *climate response to forcing is given.*

**Response**

The reference has been added.

**2 Page 2, Line 24**

*"Intuitive" is not really a good world. Our visual impression and the way we interpret it is far from being in any sense objective.*
10 *I understand what the authors say, but I kindly ask to re-formulate.*

**Response**

We agree with you.

**Modification (Page 2, line 16)**

"Although it is intuitive from Figure 1 that..." has been replaced by "Panels a and b in Figure 1 are visually very similar,
15 whereas Panel c cannot be deduced from a trivial transformation of the first panels. Therefore, it is expected that $\mu$ is "closer" to $\nu$ than $\xi$.".

**3 Page 4, line 18**

*The construction of the pullback attractor requires the integrations started at a $t = t_0$, with $t_0$ going to minus $\infty$. Otherwise no well-posed definition is possible. This should be clearly explained. Is one year of integration enough, in this case?*

**Response**

In Section 3, the integration is performed between $0$ and $\tau$. Because the dynamic of Lorenz 84 for winter and summer does not depend on time, it is equivalent to an integration between $-\tau$ and $0$. The parameter $\tau$ is chosen to be 5 "years", but we checked than 1 year is enough. Following Drótos et al. (2015), we keep $\tau$ at 5 years. In Section 4, a first integration is performed between $0$ and $\tau$, and then we integrate between $0$ and $200 \times 73$ from first integration. Due to cyclicity of seasonal forcing, the first integration is equivalent to an integration between $-\tau$ and $0$.

**Modification (Page 4, line 16)**

The parameter $\tau$ has been set at one year ($\tau = 73$), and the first integration is performed during $5\tau$ (i.e. 5 cycles / years). We have added the sentence "Snapshot attractors are special cases of *pullback attractors* (Chekroun et al., 2011). The latter class requires an integration between $-\infty$ and a desired final time. Eq. (2) does not depend of time, so the integration into Sec. 3.1 can be performed on any length intervals".

**4  Page 5, line 11**

*In this part there is no mention of the way $A$ is chosen. This seems quite important for the rest of the paper.*

**Response**

See Question 6.

**5  Page 4, line 13**

*The authors might want to note explicitly that each of the realised estimate of the measure supported by the pullback attractor come from initial conditions at $t_0$ (see point 3) distributed uniformly according to Lebesgue of the union of the little cubes.*

**Response**

We agree with you

**Modification (Page 4, line 11)**

We have replaced "we draw $N$ random initial conditions in a cube that includes the attractors, and iterate the dynamics of the systems for a time $\tau$", by "we draw $N$ random initial conditions following a uniform distribution. All margins are independent. This approximates a Lebesgue measure in a cube that includes the attractors. We iterate the dynamics of the systems between $t_0 = 0$ and a long time multiple of $\tau$."

**6 Page 6, Section 3.3**

*Discussion on the value of A is missing*

**Response**

$A$ was a misleading parameter. It represents only the number of boxes with non zero mass, but it can be different for different realization of same attractor. The value of $0.1$ gives $40$ to $60$ bins on each axis, assuming that the attractor lives in a box of $[-1;3] \times [-3;3] \times [-3;3]$. This means that the volume is divided into $40 \times 60 \times 60$ boxes. This number is the same order of magnitude as the number of gridcells in the NCEP reanalysis around the North Atlantic region, should one be interested in the climate attractor of that region (e.g. Faranda D. et al. (2017)). We also tried values of $0.05$, $0.2$ and $1.0$ for the size of the boxes (so, a factor $2$ for the two first, and one scale up for the last) for the protocol of Section 3. For all values, the maximal variation of standard deviation is $0.01$, and the detection is not affected. For a size of $0.05$ and $0.2$ the maximal variation of median is $0.03$. For the size $1.0$, the maximal increases of median of box plot of winter (resp. summer) against itself is $0.22$ (resp. $0.18$), but the difference with median of winter against summer is at least equal to $0.3$.

**Modification (Page 5, line 5 and Page 7, line 15)**

We have been added the sentence: "We chose a bin length of $0.1$ for the Lorenz attractor, which remains in a $[-1;3] \times [-3;3] \times [-3;3]$ box. Therefore $40 \times 60 \times 60$ bins cover the attractor. This number of bins is comparable to the number of gridcells that cover the North Atlantic region in the NCEP reanalysis (or most CMIP5 model simulations). This example refers to a few papers dealing with climate attractor properties (e.g. Corti S. et al. (1999); Faranda D. et al. (2017)).

The sentence "This protocol was also applied for bin sizes of $0.05$, $0.2$ and $1.0$. For $0.05$ and $0.2$, the maximal variation of median (resp. standard deviation) of Wasserstein distances is $0.03$ (resp. $0.01$), so the distributions are indistinguishable in practice. For a bin size of $1.0$, the maximal increase of the median is $0.22$, but the difference with the median of winter against summer is at least equal to $0.3$." has been added at the end of Sec. 3.3.

**7 Page 7, line 3**

*I disagree with the use of "visual impression".*

**Response**

We agree with you.

**Modification (page 7, line 11)**

"This visual impression is confirmed..." has been replaced by "This discrimination is confirmed..."

**8 Page 8, lines 13-14**

*The statement is indeed overblown if given in all generality as here.*

**Response**

We agree with you.

**5 Modification (page 8, line 25)**

"We conclude that the Wasserstein distance has a high capacity of discriminating between different dynamical systems" has been replaced by "We conclude that the Wasserstein distance has a high capacity of discriminating different attractors coming from this dynamical system"

**9 Page 9 - End of Section 4**

10 *There is a fundamental misunderstanding here, I believe. It is true that a much lower number of integrations is needed to say that two attractors are different. This is a very interesting result. But you are not able to quantify well what is (quantitatively) the difference between the expectation value of any given (possibly interesting) observable of relevance. So, you are left with a statement that is in fact qualitative rather than quantitative (the two attractors are different!). How can you relate the Wasserstein measure to any useful information? This does NOT diminish the relevance of the performed analysis, to be clear.*

15 **Response**

It is true we do not use the link between Wasserstein distance and other dynamical information to measure qualitative changes in attractors (e.g. dimensions). The chapter 9 of Villani (2003) gives some link between the Wasserstein distance and entropy.

**Modification (Conclusion)**

A caveat of the approach we present here is that we do not give an interpretation of the Wasserstein distance in terms of 20 qualitative dynamical changes (e.g. changes in local dimensions (Faranda D. et al., 2017)). Villani (2003, Chapter 9) provides links between the Wasserstein distance and entropy, but they are hard to interpret and infer for the problem we tried to tackle (measure a change in a strange attractor).

**10 Page 11, line 7**

*not clear the relationship between $\rho$ and $\mu$.*

**Response**

$\rho$ is the density of the measure $\mu$.

**Modification (page 11, line 25)**

We state that: $\rho(\geq 0)$ is the density of the measure $\mu$. Hence, for all Borel set $A$ in phase space, they are related by:

[revised manuscript text omitted]